# SheepInst: A High-Performance Instance Segmentation of Sheep Images Based on Deep Learning

**DOI:** 10.3390/ani13081338

**Published:** 2023-04-13

**Authors:** Hongke Zhao, Rui Mao, Mei Li, Bin Li, Meili Wang

**Affiliations:** 1College of Information Engineering, Northwest A&F University, Yangling 712100, China; 2Key Laboratory of Agricultural Internet of Things, Ministry of Agriculture, Yangling 712100, China; 3Shaanxi Key Laboratory of Agricultural Information Perception and Intelligent Service, Yangling 712100, China; 4Intelligent Equipment Research Center, Beijing Academy of Agriculture and Forestry Sciences, Beijing 100097, China

**Keywords:** precision livestock farming, deep learning, sheep instance segmentation, attention mechanism, computer vision

## Abstract

**Simple Summary:**

With the development of computer vision, more work is applied to promote precision livestock farming. Due to the high overlap and irregular contours of sheep, it poses a challenge to computer vision tasks. Instance segmentation can simultaneously locate and segment individuals in a sheep flock, which can effectively solve the above problems. This paper proposed a two-stage high-performance instance segmentation model, which can accurately locate and segment sheep. Under the topic of precision livestock farming, this study can provide technical support for the implementation of sheep intelligent management based on deep learning.

**Abstract:**

Sheep detection and segmentation will play a crucial role in promoting the implementation of precision livestock farming in the future. In sheep farms, the characteristics of sheep that have the tendency to congregate and irregular contours cause difficulties for computer vision tasks, such as individual identification, behavior recognition, and weight estimation of sheep. Sheep instance segmentation is one of the methods that can mitigate the difficulties associated with locating and extracting different individuals from the same category. To improve the accuracy of extracting individual sheep locations and contours in the case of multiple sheep overlap, this paper proposed two-stage sheep instance segmentation SheepInst based on the Mask R-CNN framework, more specifically, RefineMask. Firstly, an improved backbone network ConvNeXt-E was proposed to extract sheep features. Secondly, we improved the structure of the two-stage object detector Dynamic R-CNN to precisely locate highly overlapping sheep. Finally, we enhanced the segmentation network of RefineMask by adding spatial attention modules to accurately segment irregular contours of sheep. SheepInst achieves 89.1%, 91.3%, and 79.5% in box AP, mask AP, and boundary AP metric on the test set, respectively. The extensive experiments show that SheepInst is more suitable for sheep instance segmentation and has excellent performance.

## 1. Introduction

With the growth of the world’s population, people’s demand for food has gradually increased, especially meat, milk, and eggs. As a result, there is a demand for increasing the production of livestock. In recent years, the importance of Precision Livestock Farming (PLF) has grown globally. PLF is a system for monitoring and managing livestock production that focuses on improving animal welfare and optimizing livestock production [1]. In the context of PLF, it is the trend to apply artificial intelligence to improve breeding efficiency and to reduce costs [2].

The computer visual tasks are the most fundamental part in achieving efficient livestock management, and many works based on deep learning are currently proposed for PLF for different livestock. PLF manages animals as individuals, and individual identification is the foundation of any management activity [3]. Hu et al. [4] applied the YOLO model to detect cows in images and then achieved cow identification using a convolutional neural network and a support vector machine (SVM). Similarly, Shang et al. [5] used the Single Shot Detection (SSD) network to preprocess data set and designed a loss function consisting of Triplet Loss and Label Smoothing Cross-Entropy Loss function to identify sheep. In these approaches, a simple detection network was used to locate the targets before identification was performed. However, there are many interference factors in natural conditions, such as obstacle occlusion, target overlap, etc., which are not conducive to individual identification, as shown in Figure 1a.

In order to further promote the development of PLF, behavior recognition, weight estimation, and other research based on deep learning have been proposed. To reduce the interference caused by noise and occlusion, it is common to use detection and segmentation of sheep to accomplish these tasks. Yang et al. [6] implemented fully convolutional networks (FCNs) to segment still images to extract spatial features and motion analysis techniques in spatio-temporal video to identify active and inactive behaviors of sows in loose pens. He et al. [7] successively used detection and semantic segmentation to obtain more effective sheep region data for sheep weight estimation. The authors [8] used the detection and segmentation algorithm to remove the background and to extract pig features, and finally, neural networks were used to estimate pig weight. However, since the segmentation algorithm is simple, its impact on the estimated results cannot be ignored.

Each of these works has a common feature, which is to obtain the features needed for research through simple detection or segmentation. The effectiveness of image segmentation directly affects the accuracy of feature extraction and computer vision tasks [9]. It is crucial and necessary to reduce the interference of external factors and to improve the ability to perform visual tasks. High-quality image segmentation can significantly mitigate the dilemma of downstream tasks, such as individual identification, behavior recognition, and weight estimation.

Image segmentation is divided into semantic segmentation [10,11], instance segmentation [12,13,14], and panoptic segmentation [15,16]. The study objects in livestock farming usually belong to the same category, which means that instance segmentation is more appropriate for livestock farming than semantic segmentation because it can extract the location and contour of each object belonging to the same category without background interference. The performance of one-stage instance segmentation in the livestock farming scenario is unsatisfied due to the large number of highly overlapping instances and noisy background. Two-stage instance segmentation can detect potential targets before segmentation, allowing them to identify the location and size of objects more accurately in the image before segmenting each object. Due to the approximate boundaries between objects generated by detection, the two-stage method can better separate objects in complex scenes where multiple instances overlap or occlude each other compared to the one-stage method. Therefore, we examined the two-stage high-performance instance segmentation network to solve the difficulty of accurate segmentation caused by the irregular fleece and highly overlapping sheep.

Most of the current instance segmentation work is based on a two-stage pipeline of Mask R-CNN [12]. Mask RCNN is implemented by adding full convolution segmentation branches on Faster R-CNN [17], which first extracts multi-scale features by backbone and Feature Pyramid Network (FPN) [18], and then it obtains ROI (region of interest) features for the first stage to classify the target and position regression, and finally it performs the second stage of full convolution segmentation to obtain mask. Qiao et al. [19] proposed an instance segmentation method based on Mask R-CNN deep learning framework for solving the problem of cattle segmentation and contour extraction in the real environment. The authors [20] proposed the instance segmentation with Mask R-CNN of dairy cows to analyze dairy cattle herd activity in a multi-camera setting. Dohmen et al. [21] applied Mask R-CNN algorithm to segment the regions of heifers in the images to support body mass prediction. Xu et al. [22] achieved sheep behavior recognition by Mask R-CNN. Moreover, there are also some works that employed segmentation to accomplish the computer vision task of sheep [23,24].

As we see, these studies related to instance segmentation are based on the algorithm of Mask R-CNN and the edges of their research targets are relatively smooth, such as pigs, cattle, etc., which is not satisfactory on sheep data. Since the instances in the farm have high overlap and complex shapes leading to poor segmentation, the capability of Mask R-CNN cannot obtain perfect masks. This is because the detection performance has reached the bottleneck for crowded detection in sheep farms, and a lot of detailed information is lost due to pooling operation. What is more, the resolution of the mask is at 28 × 28 in general, which is too different from the original image. By visualizing the results of Mask R-CNN (as shown in Figure 2), we found that the mask boundary has a wavy shape on our data, which is caused by insufficient information when performing repeated upsampling.

Through the analysis of practical application scenarios and data content, our work is aimed finely in locating and segmenting the contour of sheep in the working area of the farm in both indoor and outdoor environments. We found that RefineMask [25] provided an idea for the problem of loss of information and low mask resolution, which can produce high-quality segmentation results through multi-stage boundary refinement. Although RefineMask worked well compared with Mask R-CNN and other methods, we found that there are still problems with sheep data. Figure 2a shows that, when RefineMask processes images with highly overlapping instances, the network cannot clearly detect each complete instance, and the detection performance needs to be enhanced.

As can be seen Figure 2c, RefineMask cannot segment the entire instance correctly, which is due to the correct features not being noticed. Based on this work, we made the following improvements: (1) to extract more excellent features, the backbone network is replaced by ConvNeXt-E instead of ResNet [26], which is obtained by adding the Efficient Channel Attention (ECA) [27] module to ConvNeXt-T [28]; (2) changing the detector to Dynamic R-CNN [29] and adding shared convolutional layers to improve the expression of the detection network; (3) adding spatial attention module (SAM) [30] to the mask head and semantic head of RefineMask, respectively, so that the segmentation network can pay attention to effective features and suppress noise.

In this paper, we constructed a sheep instance segmentation dataset on a real farm as a basis for research. Additionally, the data augmentation method of Copy-Paste [31] was applied in the training to make full use of the image data of single sheep to enrich the dataset and improve the generalization performance of the model. A high-performance instance segmentation algorithm SheepInst, focusing on the boundary segmentation effect, was proposed for sheep data in livestock farming, which provides high-quality features for the subsequent task and proposes a solution for PLF. Moreover, we provide a valuable solution for the research and application of other sheep.

The rest of this paper is organized as follows. Section 2 describes the image collection and augmentation methods. The overview of the methods, details of the improvements, and the experimental details are also presented in this section. Section 3 introduces the evaluation metrics and the experimental results of the proposed method. Finally, Section 4 discussed the application value of our work and future research directions. In Section 5, the proposed method is summarized.

## 2. Materials and Methods

### 2.1. Dataset

#### 2.1.1. Data Collection and Labelling

In this study, the research objects are Hu sheep, from Gansu Qinghuan Sheep Breeding Co., Ltd. Typically, datasets are crucial for supervised learning based on deep learning. Therefore, in order to satisfy the diversity of our dataset, we collected the images of sheep in indoor living areas and outdoor working areas from different light conditions (e.g., natural light, strong light), perspectives (e.g., overlook, side view, front view), and distances (e.g., far, near), all of which were in the JPG format with a resolution of 1920 × 1080. The living area is the place where sheep rest and eat, and the working area is the place where sheep undergo weight estimation, vaccination, and other physical examination tasks. More specifically, the work area includes a waiting area and work passage area. Considering the real situation of sheep activities, we collected images of sheep with different numbers and different degrees of overlap.

After filtering the low-quality images, 1396 images were collected in total, which were randomly divided into the training set (838 images), the validation set (279 images), and the test set (279 images) according to the ratio of 6:2:2. The images were labeled by polygons using the Eiseg tool [32], and then a dataset was generated in Microsoft COCO format [33]. Part of the dataset is shown in Figure 1a.

#### 2.1.2. Data Augmentation

The instance segmentation task of a single sheep can be easily achieved using the previous methods. Our work is aimed to improve the segmentation performance when multiple sheep instances are gathered, and thus, we introduce Copy-Paste augmentation to make full use of the data with only a single target to increase data diversity and improve the generalization ability of the model. The Copy-Paste augmentation involves extracting instances from images and randomly pasting them onto other images, creating images with multiple sheep instances. This helps to enrich the training dataset by increasing the likelihood of including scenarios that may occur in real-life.

Moreover, several data augmentation methods are used to improve the performance of the model to handle multi-scale images and mitigate the performance degradation often caused by noise in practical applications. Standard Scale Jittering (SSJ) is an image resizing and cropping method with a range of scale of 0.8 to 1.25 of the original image size. Unlike the usual scale jittering and random resizing, we employed Large Scale Jittering (LSJ) that the range of scale is 0.1 to 2.0 of the original image size to create challenging training data and increase the network’s capability of generalizing. After that, images are randomly flipped. If images are smaller than their original size, they are padded to 1024 × 1024 with gray pixel values. Finally, the processed data are normalized. The visualization of augmented images is shown in Figure 1b.

### 2.2. SheepInst

#### 2.2.1. Overview

The SheepInst framework is based on RefineMask, which is based on Mask R-CNN, and the overview is illustrated in Figure 3. All of the work is based on our sheep data. Initially, we employ a backbone called ConvNeXt-E, a combination of the convolutional neural network ConvNeXt and ECA module to extract efficient sheep features for the subsequent network. Additionally, information-rich multi-scale features are obtained by FPN. Secondly, after the ROIAlign operation locates the features from FPN to be segmented, in the object detection branch (upper branch), we introduce Dynamic R-CNN and change the structure to improve the performance in the case of multiple overlapping objects to pinpoint the instances; in the instance segmentation branch (middle and bottom branches), we perform boundary refinement through the two branches of RefineMask: (1) inputting a feature map with the highest resolution from the FPN into a semantic head to carry out semantic segmentation, as well as promoting subsequent instance segmentation by fine-grained features; (2) the mask head combines the fine-grained features in the semantic head to predict smooth instance masks by multi-stage refinement strategies. On this basis, we add SAM to the semantic head and the mask head. By implementing SAM, the semantic head can focus on the informative regions, and the mask head is able to pay attention to important features that contribute to segmentation. In the end, the high-quality masks with smoother boundaries are obtained through the multi-stage refinement of the Semantic Fusion Module (SFM) and the Boundary-Aware Refinement (BAR) strategy in the mask head to complete the segmentation.

#### 2.2.2. Feature Acquisition with ConvNeXt-E

The development of a convolutional neural network (CNN) has promoted significant progress in deep learning. CNN has been widely used in image classification, object detection, image segmentation, and other tasks, which has greatly improved accuracy and efficiency and achieved better performance than traditional methods. However, with the application of Transformer [34] in computer vision, networks, such as Vision Transformers (ViTs) [35] and Swin Transformers (Swin-T) [36], achieved state-of-the-art performance in image tasks. Although vision Transformer (e.g., Swin-T) performs well on visual tasks, the implementation of sliding window self-attention can be expensive [37]. For feature acquisition, this paper employed the pure convolutional network ConvNeXt-T.

ConvNeXt-T improves the structure and optimizes the training strategy on ResNet-50, following the design of Swin-T, while retaining the simplicity and efficiency of the convolutional network, which has better performance and faster inference speed than Swin-T. In ConvNeXt (ConvNeXt replaces ConvNeXt-T for the following), the initial stem layer, i.e., the downsampling operations, is a 4 × 4 convolution layer with stride 4, which has a small improvement in accuracy and computation compared with ResNet. As with Swin-T, the number of blocks of the four stages of ConvNeXt is set to 3, 3, 9, and 3. In ConvNeXt block, to reduce FLOPs (floating-point operations per second), the ConvNeXt block in each stage uses grouped convolutions [38]. Although this improvement reduces FLOPs, the accuracy also decreases, as expected. To compensate for the loss in performance, the network width was increased to the same number of channels (96, 192, 384, 768) as Swin-T. ConvNeXt designs the opposite structure of ResNet block and moved the depth-separable convolution [38] forward to improve the accuracy and to reduce the FLOPs. In addition, ConvNeXt increased the 3 × 3 convolutional kernel size to 7 × 7 for best performance. For the optimization strategy, the activation function and the normalization method employed GELU [39] and Layer Normalization (LN) [40], and it reduced the use of them. AdamW [41] is used as the optimizer.

However, starting from the need for higher accuracy, we further improve the ability of the backbone network to extract features. Additionally, we found that ConvNeXt still existed concerning the possibility of improvement when adding a channel attention mechanism to ConvNeXt, which can effectively address the limitation of convolutional network in capturing long-distance information, focusing on important features and suppressing unnecessary ones.

Therefore, we propose a new backbone called ConvNeXt-E by adding an ECA module [27] to the ConvNeXt structure to provide more effective features for subsequent segmentation tasks while increasing the model complexity slightly, which improves the performance with much less computational cost compared to the original structure.

First of all, we revisit the Squeeze-and-Excitation (SE) block [42], which performs global average pooling (GAP) for each channel of the input features separately to obtain information on the channel dimension. Then, the channel information is learned by two fully-connected (FC) layers, and, finally, the channel weights from 0–1 are obtained by the sigmoid function to weight the original features. Reducing the channel size at the FC layer to reduce the computational cost, we noticed that this caused a loss of critical information. Considering the information loss caused by dimensionality reduction, the ECA module uses a one-dimensional convolution instead of the two FC layers in the SE block, which perform local cross-channel interaction without dimensionality reduction and without having lower complexity, while, at the same time, maintaining performance. Additionally, the coverage of local cross-channel interactions is determined by controlling the size of the convolution kernel. The process of obtaining the channel weights by ECA is shown as follows:(1)W=σConvkGAPX
where X∈ℝW×H×C is the input, W is a C×C matrix, GAPX is the global average pooling in channels, and Conv denotes 1-dimensional convolution. σ is a sigmoid function. k denotes the size of the convolution kernel and the coverage of local cross-channel interaction, which is determined adaptively by the number of channel dimensions. The relationship between channel dimension C and kernel size k is extended to a nonlinear function, which is calculated as follows:(2)k=|log2Cγ+bγ|odd
where |t|odd indicates the nearest odd number of t, and γ and b are set to 2 and 1.

Due to ECA, the module can avoid dimensionality reduction and perform local cross-channel interaction, which bring much less model complexity and have positive effects on learning channel attention. It is a good consideration to add it to ConvNeXt. In each stage of ConvNeXt, the ECA module is embedded after the last convolutional layer. The structure is shown in Figure 4. The ConvNeXt comprises four stages with increasing numbers of channels: 96, 192, 384, and 768. The kernel size of the one-dimensional convolutional layers dynamically adjusts to 3, 5, 5, and 5, respectively, across the stages. As described in Section 3.4.2, the model parameters do not increase, but the performance improvement is obvious compared with the original model when we embed the ECA module. ConvNeXt-E has better feature extraction capabilities, and the backbone improvement brings benefits to object detection and segmentation performance. 

#### 2.2.3. Detecting Sheep with Improved Dynamic R-CNN

Detecting sheep in real-world livestock environments is a significant challenge due to their tendency to move in groups. Additionally, two-stage instance segmentation tasks follow a detect-then-segment paradigm that requires accurate object detection to achieve high-quality segmentation results. To minimize computational costs, we first employ Dynamic R-CNN with a novel training strategy, which preserves the original network structure, while enhancing object detection capabilities.

During the training process of Faster R-CNN, the quality of the proposals slowly improves. The fixed label assignment and box regression loss function limit the learning of the network, which do not provide more effective effects for the later stage of training. To address the above issues, Dynamic R-CNN adjusts the classifier and regressor in the second stage by Dynamic Label Assignment and Dynamic SmoothL1 Loss to accommodate the quality change in proposals:

(1) Dynamic Label Assignment: for the classifier, the strategy is that Intersection-over-Union (IoU) thresholds are set at a low initial value and updated continuously during training from low to high. It can improve the quality of proposals. Faster R-CNN sets the IoU threshold value to 0.5 by default, and the Dynamic R-CNN initial IoU threshold is set to 0.4.

(2) Dynamic SmoothL1 Loss: for the regressor, the strategy is to adjust the shape of the regression loss function to fit the distribution changes of regression and to compensate for the increase in proposals. During training, the gradient of the loss function becomes larger, which is conducive to making full use of the training samples and producing high-quality regression results.

However, through experiments, we have found that the detector still cannot accurately locate overlapping sheep, and the performance of the detector did not reach the upper limit on our dataset. Increasing the depth of a convolutional network within an acceptable range leads to an increase in the number of parameters that can be learned, thereby increasing the expressive power of the network. When the depth model has multiple layers, due to the interaction between parameters belonging to different layers, the signal amplification effect becomes stronger with the increase in the number of layers. During the training of the deep model, the relative difference between the common and idiosyncratic gradient directions becomes exponentially amplified due to the positive feedback between the layers [43]. In other words, the deeper the network, the more the gradient coherence phenomenon [44] is amplified, and the network has better generalization ability.

Thus, we added convolutional layers after ROIAlign, based on Dynamic R-CNN, to improve detection accuracy and to resolve the problem of difficult detection of overlapping sheep. As shown in Figure 5b, we added two 3 × 3 convolutional layers with padding of 1 before FC layers. The output channels have remained the same at 256, and FC layers are changed from one to two in order to reduce the parameters. Adding 3 × 3 convolution layers to the network to perform feature extraction can increase local context information and receptive field, which will make features more suitable for the needs of the classification and regression tasks when they are input into the fully connected layer. This improvement has effectively improved the performance of target detection by increasing small parameters and allowing for more accurate sheep segmentation. The results are shown in Section 3.4.4.

Furthermore, as shown in Figure 5c, we kept two FC layers and added two convolutional layers, and the performance of detection indeed improved, but the segmentation in the test was not as good as our best. In Figure 5d, we also tested deeper networks by adding four convolutional layers and one FC layer, which have more parameters and medium effect on testing. The structure was rejected, owing to insufficient performance improvement and a high cost. It also demonstrated that blindly increasing the depth of the network is not the most effective way to boost performance, and excessive addition of convolutions may result in the loss of important features.

#### 2.2.4. Sheep Segmentation with RefineMask-SAM

Since sheep have the characteristic of irregular boundaries and highly overlapping, a powerful segmentation head is needed to obtain fine masks. Previous instance segmentation methods, such as Mask R-CNN and Mask Scoring R-CNN, cannot satisfy our task. In addition, our sheep dataset has high resolution, and the spatial size of instances in some images is large. Therefore, we chose the algorithm for high resolution and the quality instance segmentation algorithm RefineMask as our study baselines. To further improve the performance, SAMs are added to guide the segmentation for meaningful learning, and the final algorithm is named RefineMask-SAM.

RefineMask consists of two parts (as shown in Figure 3): a semantic head and a mask head. The features after FPN are passed to these parts for instance segmentation after ROIAlign operation. The semantic head takes the highest resolution feature map of FPN (i.e., P2 layer features) as input and extracts the semantic features with full convolution without using downsampling. It uses four 3 × 3 convolutional layers and a binary classifier to predict the probability of the foreground to generate the semantic mask. The semantic head outputs the semantic features and semantic masks with the same size in three stages and inputs them after ROIAlign to the mask head for fusion as fine-grained features. It will supplement the semantic information, such as contour, shape, and texture lost in the mask head. In the mask head, the input is the 14 × 14 feature map after ROIAlign. After generating the initial mask, the instance features, instance mask, semantic features, and semantic mask are fused through the SFM in the first stage of the mask head, and then up-sampling and amplifying the spatial size of the initial mask further generates a 28 × 28 mask. Repeat the same steps twice again to obtain higher spatial size features and masks by SFM to complete the remaining stage of mask refinement. Finally, a 112 × 112 mask is obtained. From the second stage, a BAR optimization strategy is executed, which defined a specific boundary loss to make the boundary region and ground-truth closer.

However, segmentation with full convolutional networks can only consider local information, and global information is lost. Attention mechanisms are also effective in semantic segmentation and instance segmentation. Thus, this paper improved the semantic head and mask head of RefineMask by adding SAM (as shown in Figure 3) to enhance the spatial location information of the features, respectively, so that the segmentation network can focus on effective information and the quality of the segmentation boundary can be improved.

Given an input feature map from Xi∈ℝC×W×H, SAM generates global average-pooled features Favg∈ℝ1×W×H and max-pooled features Fmax, respectively, and the spatial attention weights Ws∈ℝ1×W×H are obtained by one convolution layer and the sigmoid function after concatenating them. The equation for Ws computation is as follows:(3)WsXi=σ(ConvkFavg∘Fmax
where σ denotes the sigmoid function, and Convk and ∘ represent k×k convolution layer and concatenate operation. There are two possible values for k: 3 or 7.

In the semantic head, we added SAM after four 3 × 3 convolutional layers to provide accurate semantic features and semantic masks for SFM; in the mask head, we added SAM after two 3 × 3 convolution layers to provide better instance features for subsequent generation of initial masks and multi-stage refinement. These improvements enhanced the robustness of the network in complex livestock environments. As demonstrated in the experiments in Section 3.4.5, our improvement of RefineMask is effective. As a result of the fact that a large convolution kernel is capable of obtaining relevant information in space, which has a larger receptive field and can provide more information, a 7 × 7 convolution layer in SAM was chosen in this paper.

### 2.3. Implementation Details

All experiments are run on a server with Intel(R) Xeon(R) Platinum 8160T CPU, 24 GB NVIDIA TITAN RTX GPU, and Ubuntu 18.04 OS. We apply the MMdetection [45] framework to build the project based on Python 3.8 and PyTorch 1.7.0. The hyper-parameters of our method are set as follows. The AdamW optimizer, which adaptively adjusts the learning rate, is used in the training. The base learning rate, weight decay, beta1, and beta2 are 0.0005, 0.05, 0.9, and 0.999. The number of iterations and batch size are set to 100 epochs and 2. Moreover, the learning rate decreases at 48 and 78 epochs, respectively, which helps the network model to converge.

The loss function in this paper is a multi-task loss, defined as follows:(4)$$L_{\mathrm{multi}-\mathrm{task}}=L_{\mathrm{cls}}+L_{\mathrm{box}}+L_{\mathrm{mask}}$$
where $L_{\mathrm{cls}}$ is the same classification loss as Mask R-CNN, and $L_{\mathrm{box}}$ is the Dynamic SmoothL1 regression loss. $L_{\mathrm{mask}}$ is the same as RefineMask, which consists of average binary cross-entropy loss function and BAR loss function.

## 3. Results

### 3.1. Evaluation Metrics

#### 3.1.1. COCO Metrics

To evaluate the detection and segmentation performance of the instance segmentation algorithm, we adopt the COCO metric [33]. The COCO metric employs stricter matching criteria for calculating AP values, which enables it to more accurately measure the performance of models in real-world scenarios. Moreover, the use of multiple IoU thresholds allows for a more nuanced evaluation of algorithm performance, especially for partially occluded objects. In this paper, Mean Average Precision (mAP) in COCO metrics as the main measure of model performance. AP (also known as mAP in COCO metrics) is the average of IoU thresholds for all categories from 0.5 to 0.95, with an interval of 0.05. Our dataset has only one category. Here, AP is the average of 10 thresholds. ${\mathrm{AP}}_{50}$ represents the AP at IoU = 0.5, and ${\mathrm{AP}}_{75}$ represents the AP at IoU = 0.7, which is a more stringent index.

AP is calculated as the area between the Precision–Recall curve and the horizontal and vertical axes. In the Precision–Recall curve, Recall is the horizontal axis, and Precision is the vertical axis. Precision is calculated by the following formula:(5)$$\mathrm{Precison}=\frac{\mathrm{TP}}{\mathrm{TP}+\mathrm{FP}}$$

Additionally, Recall is calculated as follows:(6)$$\mathrm{Recall}=\frac{\mathrm{TP}}{\mathrm{TP}+\mathrm{FN}}$$
where true positive (TP) means that both the prediction value and true value are positive, false positive (FP) means that the prediction is positive, but the true value is negative, and false negative (FN) means that both the prediction value and true value are negative.

In this paper, we denote box AP as ${\mathrm{AP}}^{\mathrm{box}}$. ${\mathrm{AP}}^{\mathrm{box}}$, ${\mathrm{AP}}_{50}^{\mathrm{box}}$, and ${\mathrm{AP}}_{70}^{\mathrm{box}}$ are applied to measure the performance of detection. We also denote mask AP as ${\mathrm{AP}}^{\mathrm{mask}}$ and use ${\mathrm{AP}}^{\mathrm{mask}}$, ${\mathrm{AP}}_{50}^{\mathrm{mask}}$, and ${\mathrm{AP}}_{70}^{\mathrm{mask}}$ to measure the performance of segmentation.

#### 3.1.2. Boundary AP

To further quantify the quality of boundary segmentation effect, we adopt Boundary Average Precision (Boundary AP) [46]. Boundary AP is an instance segmentation evaluation metric based on boundary IoU, which is sensitive to boundary quality. Additionally, the mask AP in the COCO metrics is based on Mask IoU. Especially for the boundary of large objects, boundary IoU can track the change in boundary quality. Since the instances in the dataset of this paper are large, and our work is refined for segmentation boundaries, this evaluation metric can quantitatively show the performance of the model. Boundary AP is also the average of 10 IoU thresholds, as described in Section 3.1.1. The Boundary IoU is defined as:(7)$$\mathrm{BIoU}\left(B,P\right)=\frac{\left|\left(B_{d}\cap B\right)\cap \left(P_{d}\cap P\right)\right|}{\left|\left(B_{d}\cap B\right)\cup \left(P_{d}\cap P\right)\right|}$$
where B and P deonte the boundary of ground-truth mask and predict mask, respectively. $B_{d}$ and $P_{d}$ denote the set of pixels within d distance from the ground–truth mask contour and the predicted mask contour. d is the pixel width of the boundary region. In the experiments, d is set to 2% of an image diagonal according to the original paper. ${\mathrm{AP}}^{\mathrm{bou}}$ denotes boundary AP in this paper.

### 3.2. Comparison with Other Methods

To verify the efficiency and accuracy of SheepInst, we conducted experiments comparing it with other methods on our sheep dataset. The default data augmentation and hyperparameters of their original methods are used, and the iterations are also set to 100 epochs. Copy-Paste and LSJ are not used in our work when the backbones are ResNet series. After training, we select the optimal model in the validation set for testing, and the results in the test set are shown in Table 1.

When the backbones of all networks are ResNet-50, the performance of our work is higher than other methods in ${\mathrm{AP}}^{\mathrm{mask}}$ and ${\mathrm{AP}}^{\mathrm{bou}}$. The detection effect of our work is only inferior to Cascade Mask R-CNN. Compared with baseline (RefineMask), which is more effective in segmentation, our work has brought an improvement of 3.0 points in ${\mathrm{AP}}^{\mathrm{box}}$, 2.1 points in ${\mathrm{AP}}^{\mathrm{mask}}$, and 4.5 points in ${\mathrm{AP}}^{\mathrm{bou}}$. Excellent results are achieved by improving detection structure and strategy and adding the attention module in the segmentation part. When the backbones change to ResNet-101, we maintain our first place with improvement of 1.3 points in ${\mathrm{AP}}^{\mathrm{box}}$, 1.1 points in ${\mathrm{AP}}^{\mathrm{mask}}$, and 0.7 points in ${\mathrm{AP}}^{\mathrm{bou}}$. Although ResNet-101 has a complex network structure, the results of sheep instance segmentation in the livestock environment are not satisfactory. Note that whether the backbone is ResNet-50 or ResNet-101, all other methods, except RefineMask, gain low scores in ${\mathrm{AP}}^{\mathrm{bou}}$. Mask R-CNN is not as competitive as other methods on sheep data.

The full body performance of SheepInst, i.e., the method consisting of ConvNeXt-E backbone, Copy-Paste, and LSJ data augmentation, has achieved the best results in ${\mathrm{AP}}^{\mathrm{box}}$, ${\mathrm{AP}}^{\mathrm{mask}}$, and ${\mathrm{AP}}^{\mathrm{bou}}$, with the best robustness and generalization. The ${\mathrm{AP}}^{\mathrm{bou}}$ is higher than baseline (RefineMask), and the experiments prove that SheepInst has an excellent performance in all aspects and is more suitable for high-quality instance segmentation of sheep in livestock environment. We also present the inference speed measured by FPS (Frames Per Second) in Table 1. SheepInst achieved 8.2 FPS and reached the practical requirements.

### 3.3. Qualitative Results

From Figure 2, SheepInst can segment each instance well even when multiple sheep are gathered, and the boundary regions of masks are smooth and complete. The other methods are not as effective on sheep data, for example, in Figure 2b, the results from Mask R-CNN, Mask Scoring R-CNN, and Cascade Mask R-CNN performed well in the overall effect, but the details are lost, and the boundary of masks are coarse. In Figure 2a,c, RefineMask cannot clearly separate each instance, and the judgment of two instances in the same box is poor. Overall, RefineMask has a good-quality boundary, but some areas are not noticed compared with our work. In contrast, SheepInst can clearly separate the contours of two different instances. Figure 6 shows more results of our work. In summary, our method has excellent segmentation performance at different angles, distances, light, backgrounds, and counts.

### 3.4. Ablation Study

#### 3.4.1. Effectiveness of Data Augmentation

We displayed the results from the validation and test sets to compare the improvement of data augmentation on model robustness and generalizability in Table 2. We first tested the effect of SheepInst using Copy-Paste and SSJ augmentation, which improves 2.0 ${\mathrm{AP}}^{\mathrm{mask}}$ and 1.7 ${\mathrm{AP}}^{\mathrm{bou}}$ without data augmentation on the test set. This data augmentation increases the diversity of samples and enhances the sheep instance segmentation performance significantly, which is a powerful data augmentation method for our sheep dataset. Secondly, we verified the effect of SheepInst using Copy-Paste and LSJ and found that the gap between the validation and test set scores became only 0.4 ${\mathrm{AP}}^{\mathrm{mask}}$, which is much smaller than the original 1.1 ${\mathrm{AP}}^{\mathrm{mask}}$. This indicates that LSJ significantly improves the generalization ability of the model.

#### 3.4.2. Effectiveness of ConvNeXt-E

We kept the remaining network after FPN unchanged and only changed the popular backbone for comparison to demonstrate the effectiveness of ConvNeXt-E in this paper. All experiments employed Copy-Paste and LSJ for data augmentation (default for later ablation experiments unless declared). As can be seen from Table 3, the ResNet series are not as effective as other backbones in extracting features on our work in this paper. Swin-T, as the competitor of ConvNeXt, has 2.4 ${\mathrm{AP}}^{\mathrm{mask}}$ less than ConvNeXt without adding the attention module on our dataset, and the gap in ${\mathrm{AP}}^{\mathrm{bou}}$ is 3.8 points. After adding the ECA module, it is 0.7 points higher than ${\mathrm{AP}}^{\mathrm{box}}$, 0.3 points higher than ${\mathrm{AP}}^{\mathrm{mask}}$, and 0.3 points higher than ${\mathrm{AP}}^{\mathrm{bou}}$. The reason for this phenomenon is that ECA could help the backbone to focus on crucial features that support high-quality detection and segmentation.

#### 3.4.3. Effectiveness of Attention Modules in ConvNeXt

We also added other attention modules after each of the four stages of ConvNeXt for comparison, as in Section 2.2.2, to prove the effectiveness of ECA. As shown in Table 4, the SE module is the founder of channel attention, which is better and lighter than Coordinate Attention (CA) [49] on our sheep data, despite the fact that many innovative approaches have surpassed it. CA and Convolutional Block Attention Module (CBAM) [30] have more parameters but do not perform as well as ECA. The structure is obtained by fusion of ConvNeXt and ECA, and it has less parameters and higher performance gains.

#### 3.4.4. Effectiveness of Improved Structure of Detector

We only changed the structure of the detector in these experiments, and the results are shown in Table 5. Compared with the original Faster R-CNN detector, our improved Dynamic R-CNN, with two convolution layers and one FC layer, improves the ${\mathrm{AP}}^{\mathrm{box}}$ by 3.9 points, ${\mathrm{AP}}^{\mathrm{mask}}$ by 0.9 points, and ${\mathrm{AP}}^{\mathrm{bou}}$ by 1.2 points. Compared with baseline (Dynamic R-CNN), our improvements are 2.0 points, 0.7 points, and 0.8 points in three metrics, respectively. This verifies that the more accurate the detector, the higher the mask AP. Retaining the original 2 shared FC layers in the detector structure results in a slight improvement in ${\mathrm{AP}}^{\mathrm{box}}$ and ${\mathrm{AP}}^{\mathrm{bou}}$, but a slight decrease in ${\mathrm{AP}}^{\mathrm{mask}}$. When we kept the original two shared FC layers in the detector structure, with slight improvement in ${\mathrm{AP}}^{\mathrm{box}}$ and ${\mathrm{AP}}^{\mathrm{bou}}$, a slightly decrease in ${\mathrm{AP}}^{\mathrm{mask}}$ was observed instead. When changing to four shared convolution layers and one FC layer, the parameters increase by 1.18 M compared with our work, but the segmentation performance decreases. In consequence, we finally chose one shared FC layer by comparing the parameters and computational cost.

#### 3.4.5. Effectiveness of SAM in Segmentation

As shown in Table 6, compared with baseline (RefineMask), we verified the SAM with the convolution kernel size 7, which results in a gain of 0.6 points ${\mathrm{AP}}^{\mathrm{mask}}$ and 0.2 points in ${\mathrm{AP}}^{\mathrm{bou}}$, which is the best result. Then, we compared two choices of convolution kernel size in SAM. The effect of convolution kernel size 7 is 0.2 higher than that of 3 in ${\mathrm{AP}}^{\mathrm{mask}}$ and ${\mathrm{AP}}^{\mathrm{bou}}$. This is due to the fact that large convolution kernels can extract more semantic information, which helps to determine the location of the target. In addition, we also tested the Squeeze-and-attention module (SA) proposed in SANet [50] in semantic segmentation and found unsatisfactory results after experiments.

## 4. Discussion

In sheep farms, there is a trend to apply artificial intelligence technologies to achieve individual management of sheep to improve farming efficiency and to reduce labor costs. However, when deploying technologies, such as individual identification, behavior recognition, and weight estimation, it was found that it was difficult to distinguish individuals and to improve the accuracy of work. The wool of the Hu sheep is more disorganised, irregular, and exuberant compared with other sheep. Moreover, the sheep’s tendency to congregate causes significant overlap of targets and interference from different lighting and backgrounds, posing a challenge for vision tasks. There are few instance segmentation methods for Hu sheep, and there is still room for exploration and improvement. The purpose of this paper is to address these problems.

We proposed a two-stage instance segmentation, called SheepInst, based on RefineMask, to achieve high-performance detection and segmentation of sheep. Compared with the baseline (RefineMask with ResNet-50), SheepInst improved the box AP, mask AP, and boundary AP by 19.7%, 8.3%, and 10.1%, respectively. When the backbone changed to ResNet-101, SheepInst improved by 7.9%, 4%, and 3.5%, respectively. Our work maximized performance gains by adding the least cost. This benefits from the following. (1) The improved backbone can focus on crucial features that support high-performance detection and segmentation. (2) We introduced new training strategies and modified the network structure in the object detection branch to improve accuracy and to achieve high-accuracy object detection to support high-quality segmentation. (3) By adding spatial attention modules to guide the segmentation network for meaningful learning, the segmentation network can focus on effective information, and the quality of the segmentation boundary can be improved.

The extensive experiments proved SheepInst outperformed the state-of-the-art models in terms of both accuracy and generalization, and the network can focus on the correct information and suppress noise, which is more suitable for the results in the sheep farm scene. Even if there is a case of highly overlapping sheep, our work can precisely locate and output high-quality masks, ensuring the success of computer vision tasks. SheepInst provides a solution to the dilemma of using computer vision technology in artificial intelligence to achieve PLF in a real livestock environment. Certainly, the other work about sheep can use our work as a pre-processing method.

The following are the limitations of our work and also the directions for improvement in the future. (1) Since the research topics nowadays gradually tend to semi-supervised, weakly supervised and unsupervised, etc., it is a common goal to achieve good segmentation performance by using a small amount of manual data annotation. The dataset labels in this paper are all manually annotated, and we can focus on semi-supervised research in the future to obtain higher model performance with less cost of labor. (2) There are limitations to the category of sheep in the dataset. We only have one category of sheep in our sheep data, and we can add other phenotypically different species, such as cashmere goats and milk goats, in the future, to increase the generalizability of the model.

## 5. Conclusions

High-performance sheep instance segmentation is significant to the research on the visual task of PLF. In this study, we constructed an instance segmentation dataset of Hu sheep for the first time. We proposed a high-performance instance segmentation algorithm SheepInst for sheep. A new backbone ConvNeXt-E was innovatively proposed by fusing ConvNeXt and ECA module, which has a reasonable number of parameters to obtain better performance than other models, and it effectively extracts the features of sheep, laying a good foundation for detection and segmentation. Next, we modified the structure of Dynamic R-CNN to enhance the performance of the object detector, and we added a spatial attention module to the semantic head and mask head of RefineMask using spatial information to help segmentation and to obtain high-quality masks. The model finally achieves 91.3% in mask AP and has the features of anti-interference, high accuracy, and a good segmentation effect of boundary.

## Figures and Tables

**Figure 1 animals-13-01338-f001:**
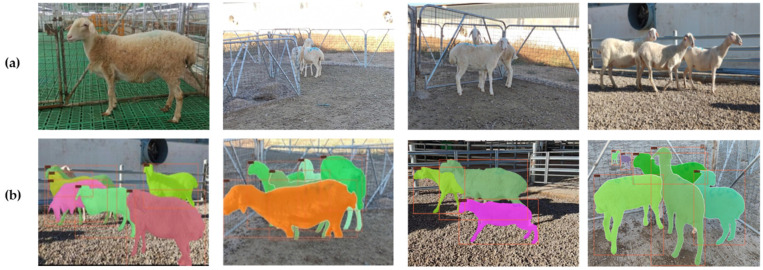
Examples of sheep images. (**a**) represents the images collected from different area, light, perspectives, and distances. (**b**) represents the training data after data augmentation.

**Figure 2 animals-13-01338-f002:**
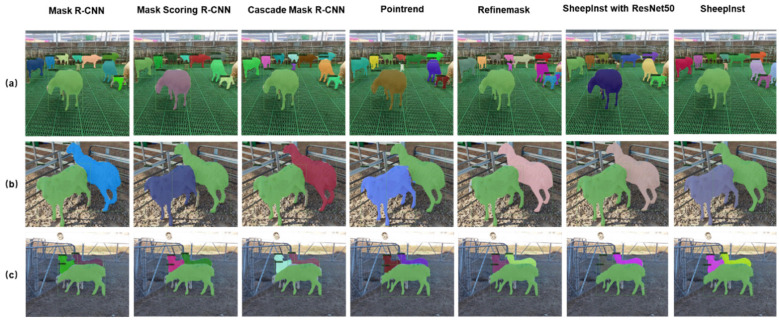
Example result from our work and other methods on test set. (**a**) represents the results of multiple sheep indoors. (**b**) represents the results of close distance outdoors (**c**) represents the results of high overlap outdoors. Except for the last column, the backbone of all the methods is ResNet-50.

**Figure 3 animals-13-01338-f003:**
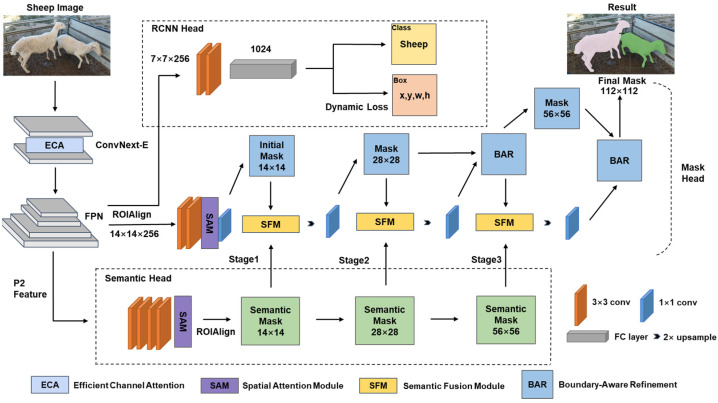
The overview of SheepInst.

**Figure 4 animals-13-01338-f004:**
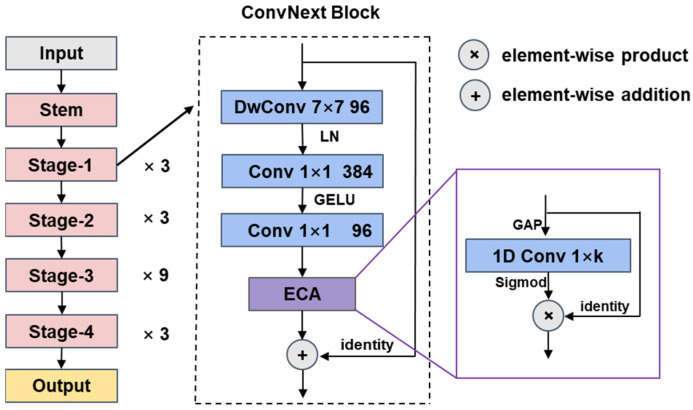
The structure of ConvNext-E.

**Figure 5 animals-13-01338-f005:**
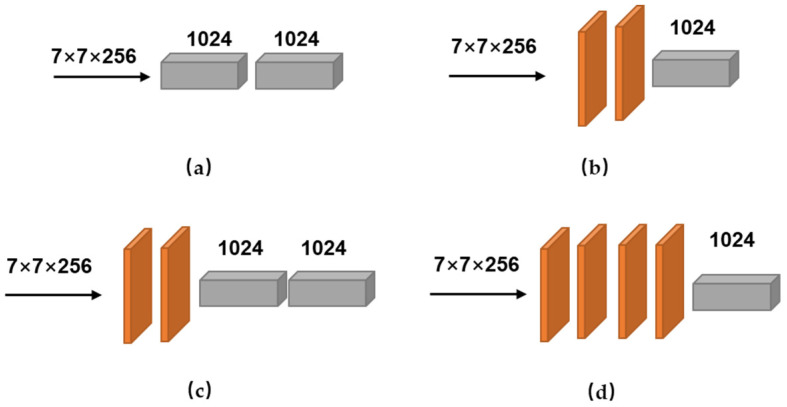
Different RCNN structures. (**a**) original RCNN structure with two FC layers. (**b**) two shared convolutional layers and one FC layer. (**c**) two shared convolutional layers and two FC layers. (**d**) four shared convolutional layers and one FC layer.

**Figure 6 animals-13-01338-f006:**
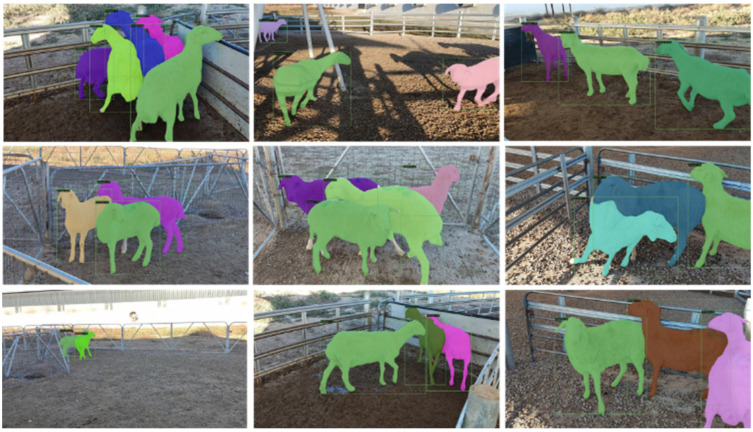
More results of our work on the test set.

**Table 1 animals-13-01338-t001:** Comparison with other methods on test set.

Methods	Backbone	${\mathrm{AP}}^{\mathrm{box}}$	${\mathrm{AP}}_{50}^{\mathrm{box}}$	${\mathrm{AP}}_{70}^{\mathrm{box}}$	${\mathrm{AP}}^{\mathrm{mask}}$	${\mathrm{AP}}_{50}^{\mathrm{mask}}$	${\mathrm{AP}}_{70}^{\mathrm{mask}}$	${\mathrm{AP}}^{\mathrm{bou}}$	FPS ^a^
Mask R-CNN [12]	ResNet-50 + FPN	77.3	93.7	88.6	75.4	93.6	87.2	55.6	12.4
Mask Scoring R-CNN [13]		77.2	93.6	86.6	75.9	92.8	86.5	55.0	12.1
Cascade Mask R-CNN [47]		81.4	82.9	87.9	74.6	90.9	85.7	54.9	10.7
PointRend [48]		78.1	93.9	87.3	82.9	93.8	89.8	70.0	11.4
RefineMask [25]		69.4	93.3	82.4	83.0	93.4	89.0	69.4	8.7
Ours w/o Aug *		78.4	93.6	88.0	85.1	93.6	89.5	73.9	7.9
Mask R-CNN	ResNet-101 + FPN	79.8	93.7	89.5	76.6	93.5	86.4	57.7	11.5
Mask Scoring R-CNN		80.0	92.8	89.7	76.5	91.8	88.7	55.7	10.6
Cascade Mask R-CNN		84.2	92.9	90.9	76.3	91.9	86.9	56.5	9.3
PointRend		81.3	92.9	89.7	84.0	93.9	90.6	71.6	9.8
RefineMask		81.2	92.8	89.8	87.3	93.7	91.6	76.0	8.6
Ours w/o Aug *		82.5	93.6	90.5	88.4	93.7	91.7	76.7	7.1
Ours w/Aug *	ConvNeXt-E + FPN	**89.1**	93.9	93.9	**91.3**	94.8	93.7	**79.5**	**8.2**

* ‘Aug’ denotes the data augmentation of Copy-Paste and LSJ. ^a^ FPS represents the number of images processed by the network per second.

**Table 2 animals-13-01338-t002:** Comparison with different data augmentation on validation set and test set.

Methods	$AP_{val}^{box}$	$AP_{test}^{box}$	$AP_{val}^{mask}$	$AP_{test}^{mask}$	$AP_{val}^{bou}$	$AP_{test}^{bou}$
SheepInst w/CP *	86.9	87.2	89.8	88.9	79.3	78.2
SheepInst w/CP&SSJ	89.0	88.6	91.7	90.8	80.6	79.8
SheepInst w/CP&LSJ	89.8	89.1	91.8	91.3	80.8	79.5

* ‘CP’ denotes the data augmentation of Copy-Paste.

**Table 3 animals-13-01338-t003:** SheepInst with different backbones on test set.

Backbone	$AP^{box}$	$AP^{mask}$	$AP^{bou}$	Parameters *
ResNet-50	85.1	88.8	77.5	23.283 M
ResNet-101	86.8	89.6	78.3	42.275 M
Swin-T	82.3	88.6	75.7	27.497 M
ResNeXt-50	85.3	88.8	76.9	22.765 M
ResNeXt-101	87.7	90.0	79.0	41.913 M
ConvNeXt	88.4	91.0	79.2	27.797 M
ConvNeXt-E (ours)	89.1	91.3	79.5	27.797 M

* Parameters are the size of the input image resolution of 1280 × 1024.

**Table 4 animals-13-01338-t004:** Comparison with different attention modules in ConvNeXt.

Backbone	$AP^{box}$	$AP^{mask}$	$AP^{bou}$	Parameters *
ConvNeXt	88.4	91.0	79.2	27.797 M
ConvNeXt + SE	88.6	91.0	79.2	27.899 M
ConvNeXt + CA	88.5	90.7	78.9	28.122 M
ConvNeXt + CBAM	88.7	91.2	79.3	28.203 M
ConvNeXt + ECA (ours)	89.1	91.3	79.5	27.797 M

* Parameters are the size of the input image resolution of 1280 × 1024.

**Table 5 animals-13-01338-t005:** Comparison with different detector structures on the test set.

Backbone	RCNN Structure	$AP^{box}$	$AP^{mask}$	$AP^{bou}$	Parameters *
Faster R-CNN	2FC	85.2	90.4	78.2	-
Dynamic R-CNN	2FC	87.1	90.6	78.7	-
Improved Dynamic R-CNN	2Conv + 2FC	89.3	91.2	79.7	+1.18 M
	4Conv + 1FC	88.6	91.0	79.4	+1.31 M
	2Conv + 1FC (ours)	89.1	91.3	79.5	+0.13 M

* Parameters are the size of the input image resolution of 1280 × 1024. ‘Conv’ denotes the convolution layer.

**Table 6 animals-13-01338-t006:** Comparison with different kernel size of SAM and other module.

Methods	Kernel Size	$AP^{mask}$	$AP_{50}^{mask}$	$AP_{70}^{mask}$	$AP^{bou}$
RefineMask	-	90.7	94.9	93.6	79.3
+SA	-	90.8	94.9	93.7	78.9
+SAM	3	91.1	94.9	92.9	79.3
+SAM (ours)	7	91.3	94.8	93.7	79.5

## Data Availability

The data presented in this study are available upon request from the corresponding author.

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
