# Peer review of "SheepInst: A High-Performance Instance Segmentation of Sheep Images Based on Deep Learning"

_animals, 2023, doi:10.3390/ani13081338_

Round 1

Reviewer 1 Report

The article aims to identify and quantify the number of sheep using deep learning. The topic is relevant, the text is well written, the article is well structured and the figures are appropriate.

Comments

The article published by Carvalho et al. (2022) developed a methodology for vehicle detection that used semantic and instance segmentation compared with the results of this article. Instead of, just considered heads, could be considered the entire body. This method is divided into four steps: semantic prediction, semantic prediction with bounndaries, instance segmentation by removing the boundaries, and restore the original size while maintaining distinct predictions. When removing the borders, the bodies become isolated, allowing for unique object identification.

- I disagreed with the authors used pure convolutional network ConvNeXt-T instead sliding window. I think an appropriate pixel Steps would give better results.

O. L. F. de Carvalho et al., "Bounding Box-Free Instance Segmentation Using Semi-Supervised Iterative Learning for Vehicle Detection," in IEEE Journal of Selected Topics in Applied Earth Observations and Remote Sensing, vol. 15, pp. 3403-3420, 2022, doi: 10.1109/JSTARS.2022.3169128.

Reviewer 2 Report

In this paper, the Authors, present an algorithmic approach which can be used to segment sheep instances in images.

The document is mostly well written, with a few English language issues. It is organized as follows: Abstract, a first Section containing an Introduction, a second Section on Materials and Methods, a third Section on the Results, a fourth Section with the Discussion, a fifth Section containing the Conclusions of the work, and finally the References used.

However, after studying the document I could not understand what is the novelty of the method, neither why it is better or worse than other alternative proposals. A detailed Discussion should be presented, comparing this proposal with others used in the area of study for segmenting animals, especially sheep. In my honest opinion, it seems to be just another segmentation approach, with no particular value to it. Please, see my comments below.

1.  In the Introduction section, please, introduce and define the concept of 'precision livestock farming'.

2.  How is currently being applied the concept of precision livestock farming in the area of sheep farming?

3.  What have people being doing in this area for tracking sheep? How did other people segment sheep? The Introduction section should also reflect this analysis.

4.  In Line 54 the needs of livestock farming are mentioned. Could you elaborate more in this line? What are their needs? How have they been collected? I recommend to add a detailed description of each of those needs.

5.  Please, in Lines 68 and 69, introduce and detail the different segmentation approaches.

6.  What was the criteria for selecting the proposed network? See Lines 75–77, and justify how can this network deal with the difficulties.

7.  What is the novelty of this proposal? Why is it better than any other state of the art algorithm? I consider that Lines 124–128 should be revised. Maybe, your proposal outperforms state-of-the-art models in this case. Maybe generalizing is not a good idea, as you have not proved that.

8.  In relation with the augmentation technique used, why have the authors decided to use Copy-Paste augmentation and no another approach? What are its benefits?

9.  I am confused, as in the Introduction section you mention some features about sheep (when the needs of livestock farming were mentioned). How is it possible to extract those features? Or are they not present in the dataset? Is the aim of this paper only related to segmentation? If that’s the case, how could you extract the features of sheep in the future?

10.  In relation with Section 2.2.1, when you mention RefineMask and Mask R-CNN, why did you choose to use this and not other? What are its advantages? Authors should introduce it.

11.  How did the Authors decide the depth of the network? Check Section 2.2.3. There is a moment when even if you increase the depth of the network, you will not get better results. What was the criteria used?

12.  Please, introduce the COCO metric and justify its choice.

13.  The Discussion section must be revised and expanded. The current Discussion is not properly discussing. What are the benefits of this proposal in comparison with other proposals? Why is it better/worse? What is its novelty?

14.  What were the limitations of this study?

15. Other comments
       •  Figure 1(a) should be moved to the introduction, after mentioning it. Same for Figure 5; otherwise, it is difficult to follow the paper.
       •  In Figure 2, please identify the input and output of the process, and use a more informative description.
       •  In Equation 2, please remove the extra closing parentheses in '|log2(C))'.
       •  In Line 269, consider replacing 'which' with 'that'.
       •  Below Table 1, consider writing 'Aug' instead of Aug.
       •  Check Line 493-494; it seems to have an error as a citation might be missing.

Reviewer 3 Report

Dear author,

Bellow you will find very little corrections that you have to do in  your manuscript

P2 L57: used detection instead of made use of detection

P2 L85-86: The authors [17] proposed instead of The authors of [17] proposed

P3L100-101: is aimed finely in locating and segmenting instead of is aimed at finely locating and segmenting

P5L214: is set to (3, 3, 9, 3). In this way doesn’t make any sense. May be the author has to remove the brackets.

P6L241-242: Reducing the channel size at the FC layer to reduce the 241 computational cost, but accidentally causing the loss of critical information needs to be rewritten to Reducing the channel size at the FC layer to reduce the computational cost, we noticed that this caused loss of critical information.

P7L261-263: Since the number of channels in the four stages of Con-261 vNeXt is (96, 192, 384, 768), the 1-dimensional convolutional kernel size k adaptively 262 changes to (3,5,5,5). The sentence has to be rewritten because English is very bad.

P7L269-274: Since sheep frequently move in groups in the real livestock environment, which 269 brings a great challenge to detection tasks. Moreover, the two-stage instance follows the 270 detect-then-segment paradigm, so we need high-accuracy object detection to support 271 high-quality segmentation. In order not to increase more computational cost, we first em-272 ploy Dynamic R-CNN to keep the original structure without increasing the network com-273 plexity but using a novel training strategy. The sentence has to be rewritten because English is very bad.

P13L508-509: When we kept the original 2 shared FC layers in detector structure, with slight improve-508 ment in APbox and APbou, and slightly decrease in APmask instead The sentence has to be rewritten because English is very bad.

Discussion: I think discussion is poor and needs to be enriched with more information’s and to be compared with other work which has been done or other methods. Need to be improved more and to refer if there is any difference or similarity with other methods or those who has used the same method in other species of sheep.

Round 2

Reviewer 2 Report

Dear Authors,

Thank you very much for the effort made in the elaboration of your detailed responses.

After revising them, I think I was finally able to properly understand the goals and the relevance of the paper.

Taking all that into consideration, I’m recommending now to accept the paper for publication.

Reviewer 3 Report

Dear author,

I would like to inform you that I have checked the revised manuscript and the author's point-by-point responses and I would like to confirm that all the necessary changes have been made. For this reason I propose that the manuscript now can be published.